# Cytotoxic alkyl-quinolones mediate surface-induced virulence in *Pseudomonas aeruginosa*

**Geoffrey D. Vrla**[1], **Mark Esposito**[1], **Chen Zhang**[2], **Yibin Kang**[1], **Mohammad R. Seyedsayamdost**[2], **Zemer Gitai**[1]*

**1** Department of Molecular Biology, Princeton University, Princeton, NJ, Unites States of America,
**2** Department of Chemistry, Princeton University, Princeton, NJ, Unites States of America

* zgitai@princeton.edu

**Data Availability Statement:** All relevant data are within the manuscript and its Supporting Information files.

## Abstract

Surface attachment, an early step in the colonization of multiple host environments, activates the virulence of the human pathogen *P. aeruginosa*. However, the downstream toxins that mediate surface-dependent *P. aeruginosa* virulence remain unclear, as do the signaling pathways that lead to their activation. Here, we demonstrate that alkyl-quinolone (AQ) secondary metabolites are rapidly induced upon surface association and act directly on host cells to cause cytotoxicity. Surface-induced AQ cytotoxicity is independent of other AQ functions like quorum sensing or PQS-specific activities like iron sequestration. We further show that packaging of AQs in outer-membrane vesicles (OMVs) increases their cytotoxicity to host cells but not their ability to stimulate downstream quorum sensing pathways in bacteria. OMVs lacking AQs are significantly less cytotoxic, suggesting these molecules play a role in OMV cytotoxicity, in addition to their previously characterized role in OMV biogenesis. AQ reporters also enabled us to dissect the signal transduction pathways downstream of the two known regulators of surface-dependent virulence, the quorum sensing receptor, LasR, and the putative mechanosensor, PilY1. Specifically, we show that PilY1 regulates surface-induced AQ production by repressing the AlgR-AlgZ two-component system. AlgR then induces RhlR, which can induce the AQ biosynthesis operon under specific conditions. These findings collectively suggest that the induction of AQs upon surface association is both necessary and sufficient to explain surface-induced *P. aeruginosa* virulence.

## Author summary

*Pseudomonas aeruginosa* is one of the most intensely studied bacterial pathogens and is a leading cause of hospital-acquired infections in the United States. An intriguing aspect of *P. aeruginosa* is its ability increase its virulence following attachment to a solid surface, suggesting that these bacteria use mechano-transduction to regulate pathogenesis. However, the cytotoxins that mediate host-cell killing in response to surface attachment remain unknown. Here, we use a microscopy-based host-cell killing assay to show that the alkyl-quinolone (AQ) family of secreted small molecules is both necessary and sufficient to explain surface-induced virulence. We further show that these compounds are

**Funding:** This work was funded by the NIH Pioneer Award DP1 AI124669-01 and the NIH NIGMS Award T32GM007388 to ZG, as well as the Princeton Catalysis Initiative to ZG and MRS. The funders had no role in study design, data collection and analysis, decision to publish, or preparation of the manuscript. There was no additional external funding received for this work.

**Competing interests:** The authors have declared that no competing interests exist.

upregulated rapidly following bacterial surface attachment and that packaging of AQs into secreted outer membrane vesicles enhances AQ cytotoxicity. This work thus fills a major gap in our understanding of surface sensing in *P. aeruginosa* and provides new methods for investigating surface-dependent signaling pathways.

## Introduction

The opportunistic human pathogen *P. aeruginosa* infects a wide range of hosts such as mammals, plants, insects, and fungi [1], and is a major contributor to the morbidity of cystic fibrosis patients [2] and hospital-acquired infections [3]. *P. aeruginosa* uses a large set of secreted proteins and secondary metabolites to carry out the multiple requirements necessary for a successful infection, including host colonization, immune evasion, nutrient acquisition, and host cell killing (cytotoxicity) [4]. Given the multiple activities involved in pathogenesis, we recently developed a quantitative imaging-based host cell killing assay to specifically study the factors acutely required for killing host cells during short timescales [5]. This assay revealed that cytotoxicity is activated by attachment of *P. aeruginosa* to a solid surface [5]. This surface-induced cytotoxicity does not require the Type-IV Pilus (TFP), TFP-associated signaling (PilA-Chp-Vfr/cAMP), or Type III Secretion Systems (T3SS), but does require two regulatory proteins, LasR and PilY1 [5]. Since well-characterized cytotoxins such as T3SS and Vfr targets are not necessary for surface-induced host-cell killing in this assay, we sought to address the outstanding questions of which specific toxins mediate host cell killing in response to surface attachment.

Understanding the pathways that act downstream of LasR and PilY1 to trigger surface-induced virulence has been particularly challenging because these regulators are known to modulate many different targets [6,7,8]. *P. aeruginosa* possesses numerous candidate toxins that could mediate surface-induced virulence, including the type III secretion system (T3SS) and numerous other secreted proteins and secondary metabolites [4]. Many of these candidates were previously found not to be required for surface-induced virulence [5], which could reflect functional redundancy or the existence of a previously overlooked cytotoxin. Furthermore, while LasR is a direct transcriptional regulator, PilY1 is not, and the two best-characterized pathways that PilY1 regulates, those dependent on cAMP and c-di-GMP [9,10], are not necessary for surface-induced virulence.

Here we characterize the pathways that activate surface-induced virulence by first showing that a single family of cytotoxins, the alkyl-quinolones (AQs), are both necessary and sufficient to explain the surface-regulated killing of *Dictyostelium discoideum* by *P. aeruginosa*, and we extend these findings to mammalian host cells. AQs had been previously known to perform multiple functions that promote virulence [11], including activating quorum-sensing pathways [12,13,14], triggering the iron-starvation response [15,16], directly targeting host-cell functions [17,18,19], and stimulating the production of cytotoxic outer-membrane vesicles (OMVs) [20]. We show that surface-induced virulence results from direct AQ cytotoxicity, as opposed to other virulence-related functions. Furthermore, we extend the role AQs play in OMV-dependent virulence by demonstrating that AQs not only stimulate OMV production, but are themselves a major cytotoxic components of these vesicles, and that vesicle packaging significantly increases the potency of AQs. Supporting their importance in surface-induced virulence, we demonstrate that surface association triggers increased AQ accumulation. Using several reporters for AQ activity we further explain how the two previously known regulators of surface-induced virulence, LasR and PilY1, influence AQ production. Together our data

indicate that surface-induced virulence results from induction of AQs, which act as toxins that directly kill host cells.

## Results

### Alkyl-quinolones are necessary and sufficient for surface-induced virulence of *P. aeruginosa* towards *D. discoideum*

Surface attachment strongly stimulates the ability of *P. aeruginosa* PA14 to kill *D. discoideum* amoebae [5]. Time-lapse imaging of *D. discoideum* exposed to planktonic *P. aeruginosa* demonstrates that *D. discoideum* completely clears the bacterial population through phagocytosis (S1 Movie). However, similar treatment of *D. discoideum* with *P. aeruginosa* that had been previously attached to a glass surface results in reduced *D. discoideum* motility followed by cell lysis (S2 Movie). This behavior is not unique to glass surfaces, as a variety of surfaces with different stiffnesses similarly induce *P. aeruginosa* virulence towards *D. discoideum* (S1 Fig and [5]). These results suggested that surface association leads to the induction of a factor (or factors) that makes *P. aeruginosa* more virulent.

To identify candidates for the toxin responsible for surface-induced virulence we used our surface-induced virulence assay to screen a number of mutants in secreted effectors or global regulatory proteins known to promote pathogenesis (S2 Fig). Specifically, we grew each mutant to the same density, allowed it to associate with a glass surface for 1 hour, added *D. discoideum* host cells, and monitored host cell death by fluorescence microscopy using the live-cell-impermeant dye, calcein-AM. Loss of many candidate *P. aeruginosa* cytotoxins, including phenazines, rhamnolipids, and hydrogen cyanide, or global regulators of virulence, including Vfr, GacA, and PvdS, did not significantly reduce surface-induced killing of *D. discoideum* (S2 Fig). In contrast, PqsA was absolutely required for surface-induced virulence (Fig 1A and S2 Fig). PqsA is an enzyme required for the biosynthesis of AQs such as PQS, HHQ, and HQNO [21], suggesting that AQs play a key role in surface-induced virulence. To determine if other factors might promote virulence at lower multiplicities of infection (MOI), we measured the killing of *D. discoideum* by wild type and *pqsA* mutant bacteria at a range of MOIs. Under no conditions did we detect significant killing by the *pqsA* mutant, while wild type bacteria effectively killed *D. discoideum* after 1 hour at MOIs as low as 50 to 1 (S3 Fig).

The AQ family of small molecules in *P. aeruginosa* performs a diverse set of virulence-related functions including quorum-sensing signaling [12,13,14], iron acquisition [15], immune suppression [17], anti-bacterial activities [18], and host cell cytotoxicity [19]. To identify the parts of the AQ pathway responsible for surface-induced virulence of *P. aeruginosa* towards *D. discoideum*, we assayed mutants deficient in the four keys steps of the AQ pathway (Fig 1B): 1) converting the anthranilate precursor to HHQ (mediated by PqsABCD), 2) converting HHQ into PQS and HQNO (by PqsH and PqsL, respectively), 3) feedback regulation onto *pqsABCDE* expression (by PQS and HHQ activating the transcriptional regulator PqsR, also known as MvfR), and 4) stimulating RhlR-dependent QS (by PqsE). Neither the production of HQNO (Δ*pqsL*) nor the activation of RhlR-dependent targets (Δ*pqsE*) was required for the killing of *D. discoideum* by *P. aeruginosa* (Fig 1C). However, *pqsA*, *pqsH* and *pqsR* mutants showed reduced ability to kill *D. discoideum* (Fig 1C).

Due to the positive feedback loop in which PqsH-produced PQS binds PqsR/MvfR (henceforth referred to as PqsR) to induce expression of the *pqsA* promoter, the reduced virulence of *pqsH* and *pqsR* mutants could be secondary consequences of reduced *pqs* operon expression. To test this possibility, we replaced the endogenous PqsR-regulated *pqsA* promoter with a strong constitutive promoter $P_{OXB20}$. This construct was sufficient to fully restore the virulence of *pqsR* and *pqsH* mutants to WT levels (Fig 1C). Constitutive expression of *pqsABCDE* does

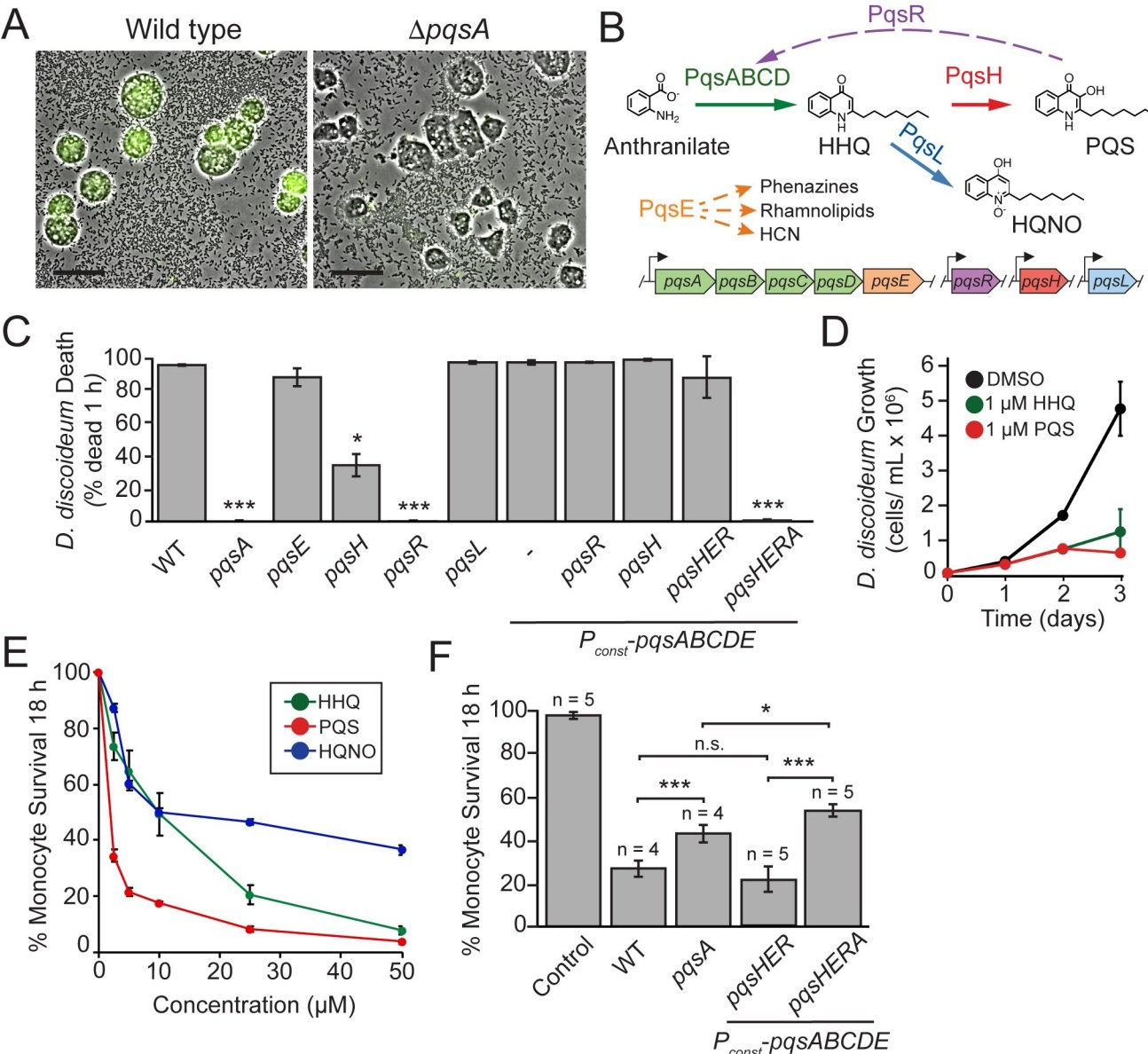

**Fig 1. Alkyl-quinolone production is necessary and sufficient for surface-induced killing of *D. discoideum* by *P. aeruginosa*.** (A) Representative images of *D. discoideum* infected with surface-attached wild type and Δ*pqsA P. aeruginosa* after 1 h co-culture (scale bars = 30 μm). Fluorescent calcein-AM staining indicates cell death. (B) Schematic of the PQS pathway depicting the functions of relevant genes. Solid and dotted arrows represent biosynthetic reactions and gene regulation, respectively. Genes above arrows indicate genetic requirements for pathways. (C) Quantification of *D. discoideum* killing by PQS pathway mutants. Expression of *pqsABCDE* is driven by the endogenous *pqsA* promoter or a strong constitutive promoter inserted upstream of the *pqsA* gene ($P_{const}$–*pqsABCDE*). Values are mean ± SEM from 3 biological replicates, and approximately 200–300 cells were analyzed for each measurement. (D) Cytotoxicity of purified HHQ and PQS to *D. discoideum* in axenic cultures grown for 3 days at 22°C. Values are mean ± SD of five biological replicates (n = 5). E) MTT cell viability assay of TIB-67 mouse monocytes after 48 h of treatment with various concentrations of the alkyl-quinolones HHQ, PQS, or HQNO in a 96-well format. Percent survival is relative to an untreated control. Values are mean ± SEM of three biological replicates (n = 3). (F) Quantification of survival of TIB-67 monocytes after 18 h co-cultured with surface-attached *P. aeruginosa*. Values are mean ± SEM of 4–5 biological replicates (n = 4–5). Approximately 2500–3000 cells were analyzed for each measurement. For statistical analysis, mutants were compared to wild type unless shown otherwise. Statistical analyses are Student's t-test (two-tailed, * = p<0.05, ** = p<0.01, *** = p<0.001).

not require extremely high expression levels, as replacing the $P_{OXB20}$ promoter with a more intermediate-strength promoter ($P_{OXB15}$) retained the ability to rescue *pqsH* virulence (though a weak constitutive promoter ($P_{OXB11}$) did not) (S4 Fig). PQS is unique among the AQs for its

ability to bind iron and in-effect trigger downstream iron-responsive virulence factors and ROS production [15,16]. Thus, the ability of constitutive *pqsABCDE* expression to restore virulence to the *pqsH* mutant suggests that these PQS-dependent signaling pathways do not control surface-induced virulence, consistent with the high virulence of a double mutant (*pvdS fpvI*) lacking two global regulators of iron-starvation response (S2 Fig). Additionally, we simultaneously deleted *pqsE*, *pqsH*, and *pqsR* in the $P_{OXB20}$-*pqsABCD* background (henceforth referred to as $P_{const}$−*pqsABCD*) and showed that these bacteria retained PqsA-dependent virulence (Fig 1C). These results suggest that surface-induced virulence requires AQs produced by PqsABCD but does not require the signaling functions associated with PqsH, PqsR, or PqsE.

The ability of expression of *pqsABCDE* to induce virulence in the absence of *pqsH* suggested that surface-induced virulence is mediated either by HHQ itself, or by another product of *pqsABCD*. To determine if HHQ is not just required, but also sufficient for killing *D. discoideum* we treated axenic *D. discoideum* cultures with commercially purified HHQ and PQS (Sigma Aldrich, St. Louis, MO). HHQ inhibited growth of *D. discoideum* at concentrations as low as 1 μM (Fig 1D). While PQS was not necessary for virulence, PQS also acted as a direct cytotoxin in this assay as purified PQS killed *D. discoideum* (Fig 1D).

## AQ cytotoxicity promotes surface-induced virulence towards mammalian cells

To support our model that surface-induced virulence is mediated by AQs, we sought to confirm that AQs can also act as direct cytotoxins towards a relevant mammalian cell line. Specifically, we added various concentrations of each of HHQ and PQS to TIB-67 mouse monocyte cells under standard culture conditions in a 96-well format and assessed viability after 48 hours using a water-soluble tetrazolium assay (Fig 1E). Both HHQ and PQS inhibited growth of TIB-67 monocytes, with $LD_{50}$ values of approximately 10 μM and 1 μM, respectively (Fig 1E).

We next utilized a monocyte infection assay (S5 Fig) to determine if AQ production influences the ability of *P. aeruginosa* to kill monocytes. Treatment of monocytes with surface-attached WT *P. aeruginosa* resulted in significantly more monocyte death compared to treatment with the *pqsA* mutant (Fig 1F). To more precisely distinguish the effect of AQ cytotoxicity from other activities mediated by AQs, we assayed the virulence of *pqsH pqsE pqsR* $P_{const}$−*pqsABCDE* and an isogenic, *pqsA* null, derivative (Fig 1F and S6 Fig). The strain producing HHQ caused significantly more killing than the *pqsA* mutant (Fig 1F and Fig S6), indicating that PQS-independent AQ activity increases the cytotoxicity of *P. aeruginosa* towards TIB-67 mouse monocytes. The fact that loss of AQ production completely eliminates surface-dependent virulence towards amoebae (Fig 1C) but has an intermediate effect on surface-dependent virulence towards monocytes (Fig 1F) indicates that there are also additional factors that promote cytotoxicity in context of monocyte infection.

The AQ concentrations observed under standard *P. aeruginosa* culture conditions and at multiple *P. aeruginosa* infection sites have been reported previously [17,22,23,24,25,26]. These reports show that HHQ concentrations can reach as high as 20 μM under some culture conditions [26], but are more often found to average ~7.5 μM, depending on the conditions and timescale analyzed [17,22,23,24,25]. These values are within the range or close to the HHQ concentrations that we show are sufficient to substantially inhibit the growth of *D. discoideum* (MIC = 1 μM) and monocytes ($LD_{50}$ = 10 μM). To determine the concentration of AQs capable of causing the rapid death on the short timescales (< 1 hour) of the *D. discoideum* killing assay, we placed *D. discoideum* under agar pads supplemented with purified HHQ and measured cell death at 1 hour (S7 Fig). Within one hour, complete killing was observed following

treatment with 50 μM HHQ (similar to the killing achieved by WT *P. aeruginosa*), while 40 μM HHQ killed approximately 50% of the *D. discoideum* (similar to the killing achieved by *pqsH* mutants). Thus, the levels of purified HHQ required to kill *D. discoideum* are significantly higher than the levels reported to be typically produced by *P. aeruginosa* [17,22,23,24,25,26]. This suggests that surface attachment may strongly increase AQ levels beyond the concentrations previously reported. Alternatively, AQs produced endogenously by the bacteria could be more potent or synergistic with additional virulence factors than purified AQs delivered exogenously.

## An AQ biosensor reveals that surface-association leads to increased levels of AQs

To address the discrepancy between the levels of purified AQs required for host killing and the levels typically produced in planktonically grown *P. aeruginosa* cultures, we first sought to quantify the extent to which surface attachment increases AQ levels. AQ quantification under the conditions of our surface-induced virulence assay is challenging using traditional MS-based techniques. Consequently, we developed a fluorescence-based AQ biosensor that can be used to measure PQS and HHQ levels in surface-attached bacteria. Specifically, we engineered a reporter strain with three features: 1) it is unable to synthesize AQs itself (due to deletion of *pqsA*), 2) it linearly responds to PqsR activation without quorum-sensing feedback (due to replacement of the *pqsR* promoter with a constitutive $P_{tac}$ promoter), and 3) it has a plasmid containing both a fluorescent YFP reporter for PqsR activation by AQs ($P_{pqsA}-YFP$) and a constitutive mKate reporter ($P_{rpoD}-mKate$) to normalize for plasmid copy number.

We validated our AQ biosensor by analyzing its response to purified AQ standards (Sigma-Aldrich, St. Louis, MO) in a 96-well format. The AQ biosensor exhibited a dynamic range with a detection threshold of 0.15 μM for PQS and 2.5 μM for HHQ and a saturation level around 100 μM for PQS and 50 μM for HHQ in the plate-based assay (Fig 2A). Consistent with the known binding affinities of PqsR [26], the sensor responded more strongly to PQS than HHQ and did not respond to HQNO (Fig 2A). To validate the ability of this biosensor to measure AQ levels using fluorescence microscopy under the conditions of the surface-induced virulence assay, we doped the AQ biosensor (1:50) into a monolayer of Δ*pqsA* bacteria (prepared as described in the *D. discoideum* killing assay) and covered them with agar pads containing various concentrations of purified HHQ (Fig 2B). The response of the biosensor strain to purified AQ standards when doped into a lawn of *P. aeruginosa* was measured by fluorescence microscopy and used to construct a standard curve (Fig 2C). Under these conditions, the biosensor responded similarly to the results using a microplate reader (Fig 2A), but with lower detection threshold of 0.1 μM HHQ (presumably due to increased sensitivity) and a linear range up to approximately 15 μM (Fig 2C). Finally, we confirmed that strains constitutively expressing low ($P_{OXB11}$), intermediate ($P_{OXB15}$), and high ($P_{OXB20}$) levels of *pqsABCDE* led to the expected increases in AQ biosensor activity (S8 Fig).

Having validated our AQ biosensor, we used it to compare AQ levels between planktonic and surface-attached *P. aeruginosa* populations. Because the AQ biosensor responds to both HHQ and PQS, we focused on quantifying AQs from Δ*pqsH*, which makes HHQ but not PQS. We note that this strain is less virulent than wild type but retains 40% of its virulence and its virulence is still specifically induced by surface-association (Fig 1C). We doped the AQ biosensor (1:50) into surface-attached and planktonic populations of Δ*pqsH* at the time of *D. discoideum* addition. The AQ biosensor is itself avirulent such that doping it at low levels (1:50) enabled us to quantify AQ production without disrupting the assay. Comparing the biosensor signal, we observed a significant increase in biosensor signal in surface-attached populations

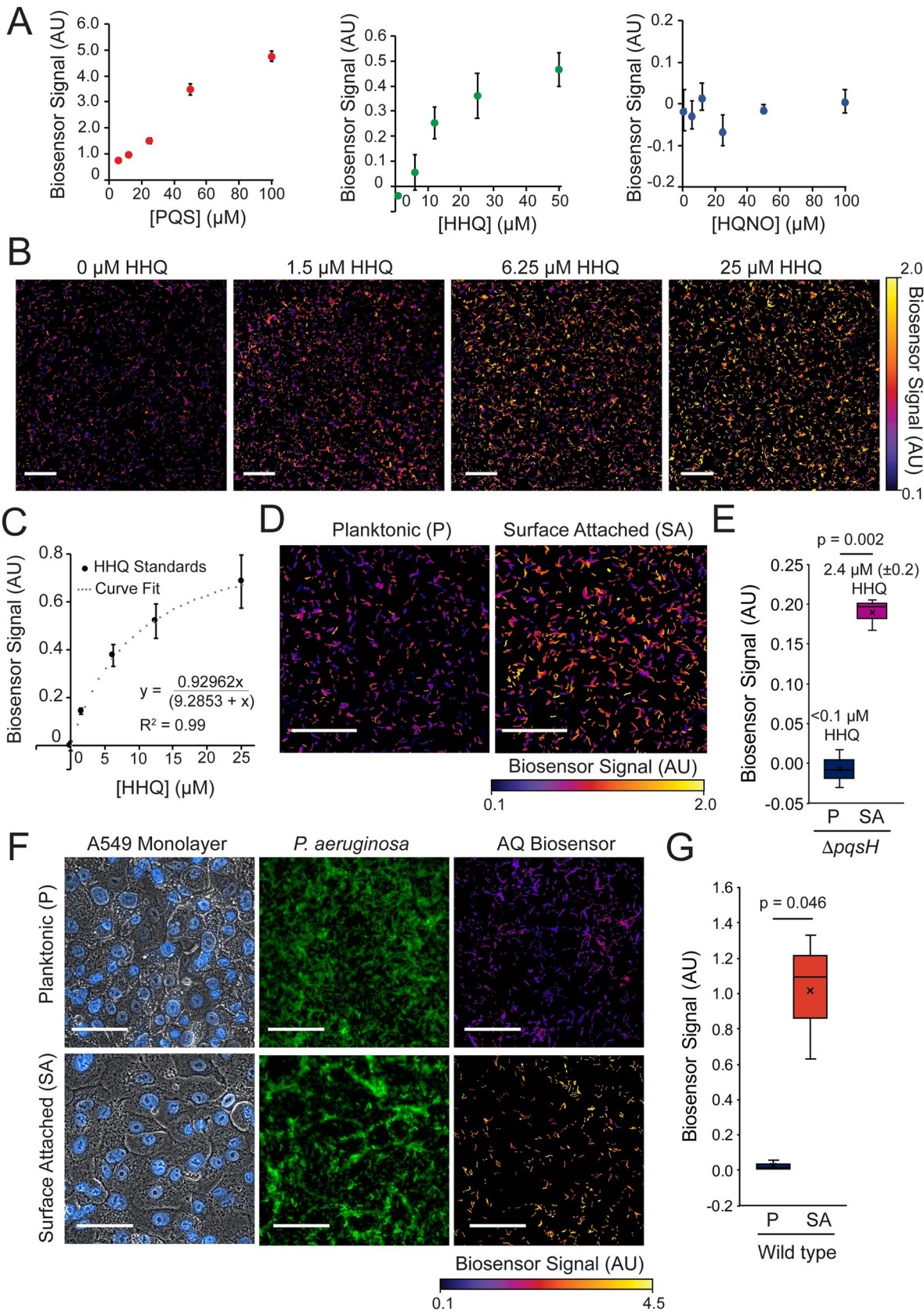

**Fig 2. Biosensor-based detection of alkyl-quinolones in surface-attached and planktonic *P. aeruginosa* populations.** (A) Response of AQ biosensor to increasing concentrations of purified PQS, HHQ, and HQNO in a 96-well microplate-based assay. Values are mean ± SD of six technical replicates (n = 6). (B) Representative images of AQ biosensor treated with various concentrations of HHQ. Biosensor cells were doped 1:50 into surface-attached Δ*pqsA P. aeruginosa*, and covered with a 1% agar pad supplemented with HHQ (scale bars = 50 μm). (C) Standard curve of the AQ biosensor response to purified HHQ in (B). Values are mean ± SD of three technical replicates (>2,500 cells analyzed per sample). (C) Representative images of AQ biosensor doped 1:50 into samples of surface-attached or planktonic Δ*pqsH P. aeruginosa* prepared following the conditions of the *D. discoideum* cell death assay (scale bars = 50 μm). (E) Quantification HHQ levels in surface-attached and planktonic *P. aeruginosa* shown in (D). Values are mean biosensor signal ± SEM for three biological replicates (>7,500 cells analyzed per sample). (F) Comparison of biosensor response when doped into wild type *P. aeruginosa* bacteria previously attached to a monolayer of A549 human lung epithelial cells for 1 h to that of planktonic bacteria immediately after exposure to an A549 monolayer. *P. aeruginosa* are expressing a constitutive GFP construct to allow visualization of bacterial populations (scale bars = 100 μm). High YFP expression in biosensor cells dope into the surface attached population resulted in high signal in the GFP channel. (G) Quantification of biosensor signal doped in planktonic and A549-attached bacteria shown in (F). Values are mean biosensor signal ± SEM for three biological replicates (>2,500 cells analyzed per sample). Cell stains (Hoescht and FM 4–64) and GFP⁺ *P. aeruginosa* were not used for experiments quantified in (G) to prevent interference with biosensor signal. Statistical tests are Student's two-tailed t-test. Biosensor signal is the mean $P_{pqsA}$-YFP / $P_{rpoD}$-mKate fluorescence per cell subtracted by this value for the DMSO (0 μM HHQ) control condition.

as compared to planktonic Δ*pqsH* populations (Fig 2D and 2E). Conversion of biosensor signal to HHQ concentration using the standard curve (Fig 2C) indicated that the HHQ concentration in surface-attached *P. aeruginosa* population during *D. discoideum* infection is 2.4 ± 0.2 μM (Fig 2E), while HHQ was undetectable in the planktonic condition (statistically indistinguishable from the 0 HHQ control) (Fig 2E). This reflects a fold increase by a factor of at least 20 and indicates that surface attachment stimulates AQ accumulation.

Given our results that association with a glass surface stimulated AQ accumulation, we wanted to determine if a biological surface such as a host-cell monolayer can also stimulate AQ accumulation. First, we confirmed the levels of bacterial attachment to a monolayer of A549 human lung epithelial cells after co-culture for 1 hour with *P. aeruginosa* constitutively expressing GFP (S9 Fig). We then used our fluorescent AQ biosensor to compare AQ accumulation in *P. aeruginosa* cells that were attached to a monolayer of A549 human lung epithelial cells for 1 hour to that of planktonic cells immediately after exposure to an A549 monolayer (Fig 2F). We observed significantly higher AQ biosensor signal when doped into populations of A549-attached bacteria (Fig 2G), suggesting that association with human cells can also stimulate AQ accumulation.

## Packaging of AQs in secreted outer membrane vesicles increases AQ cytotoxicity

One difference between bacterially produced and purified AQs is that when bacteria make AQs they are often packaged into OMVs [20]. AQs stimulate the production of secreted outer-membrane vesicles (OMV) and are abundant in OMVs themselves [20]. Furthermore, OMV's can induce rapid cell death in mammalian cell lines [27,28,29], but the specific cytotoxin responsible for this killing is unclear. To determine if OMVs are cytotoxic to *D. discoideum*, and if AQs are responsible for this cytotoxicity, we treated axenic *D. discoideum* cultures with equal amounts of OMVs purified from wild type or Δ*pqsA P. aerguinosa* and monitored *D. discoideum* death (Fig 3A). OMVs containing AQs resulted in nearly complete *D. discoideum* death after 1 hour, while no significant cell death was observed in the *D. discoideum* treated with OMVs from the *pqsA* mutant (Fig 3A). Thus, AQs are a major contributor to the cytotoxicity of OMVs towards *D. discoideum*.

We next sought to test the requirement for OMVs to be intact in order to elicit this cytotoxic response. To allow visualization of OMVs, we fluorescently tagged an outer-membrane lipoprotein that gets targeted to OMVs (*oprM*¹⁻²⁴::*mNeonGreen*), allowing us to visualize intact OMVs using fluorescence microscopy (S10 Fig and Fig 3B). We confirmed that treatment of

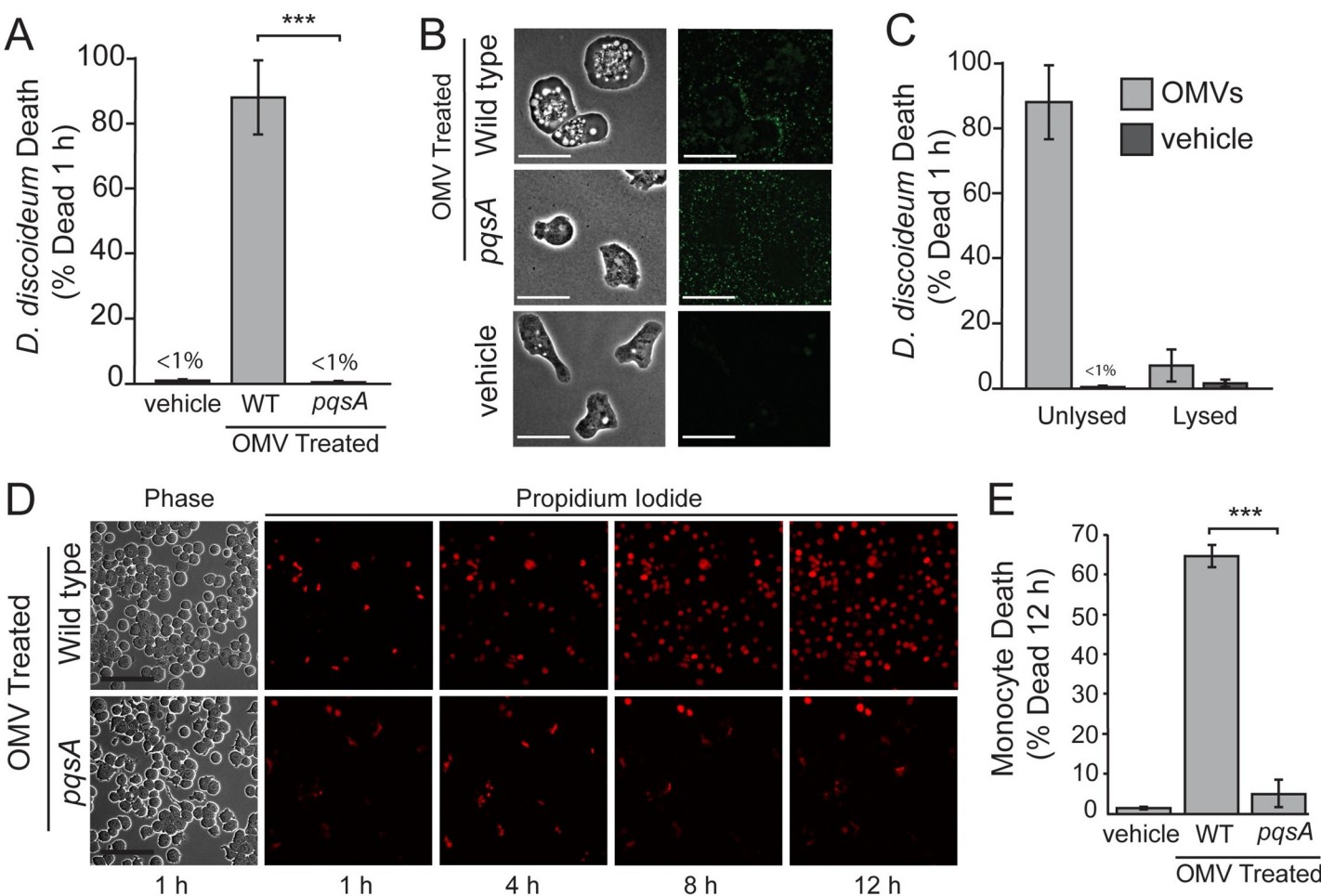

**Fig 3. AQs are responsible for OMV cytotoxicity and OMV packaging enhances AQ potency.** (A) Quantification of *D. discoideum* death following treatment of axenic cultures with outer membrane vesicles (OMVs) isolated from wild type (WT) and Δ*pqsA* mutant *P. aeruginosa* or a vehicle control (t = 1 h). (B) Images of *D. discoideum* treated with fluorescently labelled OMVs (*oprM::mNeonGreen*) from wild type and *pqsA* backgrounds in the absence of live-dead calcein-AM dye after 1 h (scales bars = 10 μm). (C) Quantification of *D. discoideum* death following treatment of axenic cultures with lysed or unlysed outer membrane vesicles (OMVs) isolated from wild type (WT) *P. aeruginosa* (t = 1 h). (D) Representative images of TIB-67 mouse monocyte at various timepoints following treatment with OMVs isolated from wild type (WT) and *pqsA* mutant *P. aeruginosa* or a vehicle control. Propidium iodide (red) staining of nuclei indicates cell death (scale bars = 100 μm). (E) Quantification of monocyte death in (D). Percent death is the increase in PI-stained nuclei divided by the total PI-negative cells at 1 h. Values are mean +/- SEM. Approximately 2500–3000 cells were analyzed for each measurement. Data shown in (B) and (C) are mean and SD of two independent experiments. Approximately 300–500 cells were analyzed for each condition. Vehicle control is solution collected by filtering OMV samples in PBS (pH = 7) through 100k MWCO filters. Statistical analysis in B and E is Student's two-tailed t-test (*** = p < 0.001).

OMVs with 0.05% SDS efficiently lysed OMVs using the fluorescently tagged strains (S10 Fig). We lysed OMVs isolated from wild type *P. aeruginosa* with 0.05% SDS, and then treated *D. discoideum* with lysed and unlysed samples in a manner such that the final assay is performed with the same level of SDS in both cases (Fig 3C). Unlysed OMVs produced greater than 10-fold more cell killing than lysed OMVs within 1 hour (Fig 3C). Given that both samples contained the same levels and composition of AQs and SDS, we conclude that the increase in cytotoxicity of the unlysed samples results from the presence of these AQs in intact OMVs. Our previous results that OMVs lacking AQs are not cytotoxic under these conditions (Fig 3A) eliminates the possibility that the loss of toxicity is a result of some other cytotoxin packaged in the OMVs being unable to be delivered to the host cell.

We next sought to determine if AQs also mediate the cytotoxicity of OMVs towards mammalian host cells. We therefore treated monocytes with equal amounts of OMVs purified from wild type or Δ*pqsA P. aerguinosa* and monitored monocyte death (Fig 3D and 3E). OMVs containing AQs were significantly more cytotoxic than those that did not (Fig 3D and 3E), indicating that AQs are also a major contributor to the cytotoxicity of OMVs to monocytes as well as *D. discoideum*.

Given that packaging of AQs in OMVs enhances the cytotoxicity of these compounds, we sought to determine if OMV packaging also influences the ability of AQs to stimulate quorum-sensing pathways through inter-bacterial signaling. To test this, we utilized the AQ biosensor, which uses the AQ quorum sensing response factor, PqsR, as its receptor. We treated the AQ biosensor strain with lysed and unlysed OMVs isolate from wild type *P. aeruginosa* and measured the biosensor response over time. There was no significant difference in the AQ biosensor response in samples of lysed and unlysed OMVs (S11 Fig), and the responses all fell within the linear range of the AQ biosensor (Fig 2A and S11 Fig). These results indicate that in contrast to cytotoxicity, AQ-based quorum sensing is not affected by whether these compounds are packaged in intact OMVs.

## AQ regulation can explain the effects of known surface-induced virulence regulators

We previously showed that LasR and PilY1 are required for surface-induced virulence [5]. To understand whether the virulence defects of these mutants are due to loss of AQ production, we determined if they can be rescued by replacing the endogenous *pqsA* promoter with the $P_{OXB20}$ strong constitutive promoter ($P_{const}$). This *pqsABCDE* overexpression restored full virulence to surface-associated *lasR* and *pilY1* deletion mutants (Fig 4A and 4B). Furthermore, constitutive *pqsABCDE* expression was sufficient to induce virulence in planktonic bacteria (Fig 4A and 4B). While the virulence achieved by expression of *pqsABCDE* in planktonic cells did not reach the same levels as the surface-attached bacteria, the conversion of avirulent cells to a virulent state with only the expression of a single operon is notable.

We next sought to determine if *pqsABCDE* expression is altered in *lasR* and *pilY1* mutants using a fluorescent reporter fusion to the *pqsABCDE* promoter. We fused a 500-bp fragment upstream of the *pqsA* gene to a promoterless *mCherry* gene and integrated this construct at a neutral chromosomal locus. Wild type, Δ*lasR*, and Δ*pilY1* bacteria expressing this $P_{pqsA}$-*mCherry* reporter were grown to mid-exponential phase (OD = 0.6), and OD-matched cultures were allowed to attach to a surface for 1 hour. We measured the mean fluorescence intensity of individual surface-attached bacteria and normalized them to a constitutively expressed fluorescent reporter. The *pqsABCDE* promoter activity of both Δ*lasR* and Δ*pilY1* was significantly lower than that of wild type (Fig 4C and 4D). To determine if the decreased promoter activity impacts AQ production, we measured HHQ and PQS levels in ethyl acetate extracts of wild type, Δ*lasR*, and Δ*pilY1* cultures grown to early stationary phase using LC/MS (OD = 1.5). We note that AQ levels have typically been measured in high-density cultures in which PqsH levels are high [22,26], such that our use of lower-density cultures in in which PqsH is relatively low can explain why we observe higher HHQ levels than PQS levels. Our early stationary phase analysis revealed that both HHQ and PQS production are decreased in both Δ*lasR* and Δ*pilY1* compared to wild type (Fig 4E). We note that the HHQ concentration determined by LC/MS (3.4 μM) is higher than that reported in Fig 2E. This is expected, since the planktonic bacteria analyzed by our biosensor were isolated from less dense cultures (OD = 0.7–0.8 compared to OD = 1.5), and they were performed on the *pqsH* mutant, which overall exhibits lower levels of *pqsABCDE* expression compared to wild type bacteria [12]. Together these results suggest that

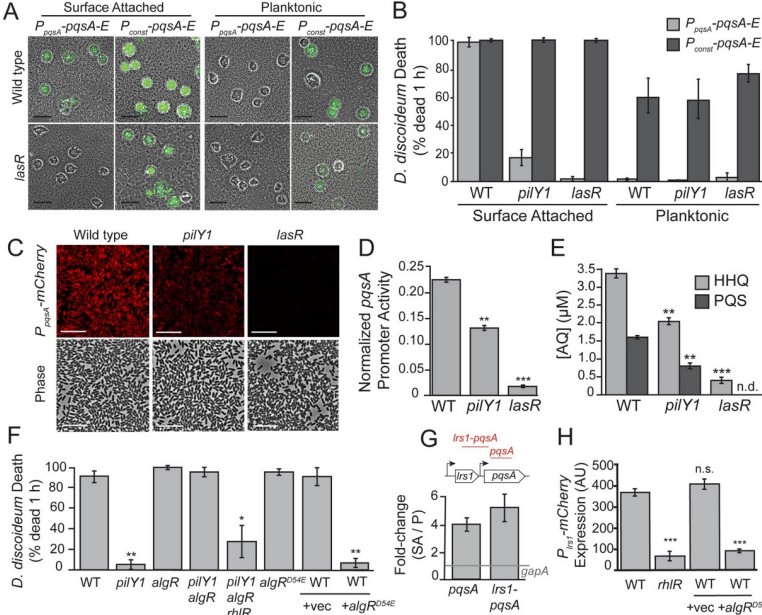

**Fig 4. PilY1 and LasR promote surface-induced virulence through *pqsABCDE* expression.** (A) Representative images of *D. discoideum* infected with surface-attached and planktonic *P. aeruginosa* expressing *pqsABCDE* genes under control of the endogenous *pqsA* promoter ($P_{pqsA}$−*pqsABCDE*) or a strong constitutive promoter ($P_{const}$−*pqsABCDE*) after 1 h of co-culture (scales bars = 50 µm). Fluorescent calcein-AM staining indicates cell death. (B) Quantification *D. discoideum* killing by surface-attached and planktonic wild type, Δ*pilY1* and Δ*lasR P. aeruginosa* after 1 h co-culture. (C) Representative images of surface-attached *P. aeruginosa* expressing a $P_{pqsA}$−*mCherry* fluorescent reporter (scale bars = 10 µm). (D) Mean fluorescence intensity per cell (>500 cells) of surface-attached *P. aeruginosa* expressing a ($P_{pqsA}$−*mCherry*) promoter fusion normalized by the expression of a constitutive $P_{tac}$-*mCherry* reporter. Values are mean ± SEM of three biological replicates (n = 3). (E) LC/MS-based quantification of HHQ and PQS in extracts of wild type, Δ*pilY1*, and Δ*lasR* liquid cultures grown to OD = 1.5. Values are mean ± SEM of three biological replicates (n = 3), and concentrations were calculated using a standard curve constructed from purified AQ standards (n.d. = not detected). (F) Quantification *D. discoideum* killing by mutants in the AlgR-PilY1 pathway after 1 h co-culture. (G) qRT-PCR analysis comparing relative transcript abundance in surface-attached to planktonic *P. aeruginosa* after 1 h growth on a surface, with diagram showing relative amplicon positions (red lines) in the *pqsA* gene and upstream regions. Values are mean ΔΔCt values ± SEM (n = 3 biological replicates) relative to the *gapA* control transcripts. (H) Mean fluorescence intensity per cell (>500 cells) of surface-attached *P. aeruginosa* expressing a ($P_{lrs1}$−*mCherry*) promoter fusion. Values are mean ± SEM of three biological replicates (n = 3). Values in (B) and (F) are mean ± SEM of three biological replicates (n = 3). Statistical analysis was performed against wild type (Student's t-test, two-tailed, * = p<0.05, ** p <0.01, ***p<0.001). Approximately 150–300 cells were analyzed for each measurement in (B) and (F).

the effects of known virulence regulators can be explained by their effects on *pqsABCDE* operon expression.

LasR has previously been shown to induce the PqsR *pqsABCDE* regulator [13], but the connection between PilY1 and *pqsABCDE* expression has not been previously reported. Since the well characterized c-di-GMP and cAMP pathways do not appear responsible for PilY1-mediated virulence induction (S2 Fig and [5]), we compared the previously reported surface-dependent transcriptional changes in WT and *pilY1* mutants [5]. We found that both the AlgR-AlgZ two-component system (TCS) and its associated regulon are repressed in a PilY1-dependent-manner [5,30]. To test if AlgR can regulate virulence we generated a phosphomimetic mutation in *algR* (*algR$^{D54E}$*) and overexpressed it from a multi-copy plasmid. AlgR$^{D54E}$ overexpression strongly inhibited virulence (Fig 4F). To determine if AlgR functions downstream of PilY1 we turned to epistasis analysis and generated a *pilY1 algR* double mutant, which restored virulence to the avirulent *pilY1* single mutant (Fig 4F). Thus, *algR* is epistatic to

*pilY1* and likely functions downstream of it. To determine if PilY1 acts on the levels or phosphorylation state of AlgR, we replaced the chromosomal copy of *algR* with a phosphomimetic *algR*$^{D54E}$ allele. Unlike in the case of overexpressing *algR*$^{D54E}$ from a plasmid, chromosomal expression of *algRD54E* did not affect virulence (Fig 4F), which suggests that PilY1 functions by transcriptionally repressing the levels of *algR*. Quantification of AQ levels in surface-attached wild type, *pilY1*, *algR*, and *pilY1 algR P. aeruginosa* confirmed this epistatic relationship (S12 Fig). While our studies indicate that AlgR functions downstream of PilY1, AlgR was also shown to regulate expression of the *pilY1* operon [30], suggesting that these proteins are involved in a feedback system.

AlgR is a transcriptional regulator but does not appear to directly regulate *pqsABCDE* expression [31], suggesting that there is another factor involved. One candidate is RhlR, as AlgR is known to repress the *rhl* QS system [32], and there are multiple predicted RhlR binding sites upstream of the *pqsABCDE* operon [13]. Consistent with the high virulence of *pilY1 algR* requiring *rhlR*, a triple deletion of *pilY1 algR* and *rhlR* led to a reduction in virulence similar to the *pilY1* single mutant (Fig 4F).

Our epistasis results suggest that RhlR positively regulates AQ-dependent surface-induced virulence. However, previous reports demonstrated that in high-density liquid cultures, *rhlR* mutants have increased expression of the *pqsABCDE* genes [13], indicating that RhlR acts to repress this operon. We recently reported that a small RNA immediately upstream of *pqsA*, *Lrs1*, is also surface induced [33] and has a RhlR binding site in its promoter. Since the *lrs1* sequence is predicted to form a secondary structure that could disrupt *pqsABCDE* promoter activity, we hypothesized that RhlR could exert either positive or negative effects on *pqsABCDE* expression depending on whether the transcript made by the *Lrs1* promoter is terminated. To test this model we first sought to determine if surface-attached *P. aeruginosa* bacteria exhibit readthrough from the *Lrs1* promoter into the *pqsABCDE* operon. We thus performed qRT-PCR on total RNA extracted from surface-attached and planktonic *P. aeruginosa* with primers both entirely within the *pqsA* coding region, and primers that span the intergenic region between *Lrs1* and the first several codons of the *pqsA* genes (*pqsA-lrs1*). Both the *pqsA* transcript and the intergenic region were upregulated in surface-attached cells compared to housekeeping control genes (Fig 4G). To test if the *Lrs1* promoter is indeed RhlR-dependent, we generated a transcriptional reporter by fusing it to *mCherry*. Deletion of *rhlR* resulted in a ~7-fold reduction in P$_{Lrs1}$-*mCherry* promoter activity (Fig 4H). Furthermore, overexpression of *algR*$^{D54E}$ resulted in a similar decrease in reporter expression (Fig 4H). Together, these results show that the *Lrs1* promoter upstream of the *pqsABCDE* operon is induced by RhlR, and can lead to increased *pqsABCDE* expression in surface-attached cells.

## Discussion

Multiple lines of evidence support our conclusion that the surface-dependent virulence of *P. aeruginosa* is mediated by induction of AQ cytotoxins. Loss of PqsABCDE, which inhibits AQ production, leads to loss of surface-induced virulence towards *D. discoideum* and significantly reduces the surface-induced virulence towards monocytes. Thus, AQ production is important for surface-induced virulence towards multiple host types. Restoring PqsABCD in the absence of PqsE, PqsH, or PqsR rescues surface-induced virulence, indicating that a PQS-independent AQ, such as HHQ, is sufficient to induce virulence even in the absence of genes responsible for AQ-mediated QS or iron-dependent signaling. PqsABCD overexpression is also sufficient to induce virulence in planktonic cells, which supports the conclusion that these cells have reduced virulence due to low AQ abundance. Finally, we confirmed that purified HHQ and PQS are sufficient to directly kill host cells in the absence of bacteria, indicating that these

factors are themselves cytotoxins as opposed to regulators of additional factors. These findings reinforce recent suggestions that AQs serve multiple important functions in virulence [20] and add surface-induced virulence factors to this growing list.

Our findings also implicate AQ production as a powerful reporter for dissecting the signal transduction pathways responsible for surface-induced virulence. Our analysis of the two known regulators of surface-induced virulence in the *Dictyostelium* cytotoxicity assay, PilY1 and LasR, showed that both regulators control virulence by activating *pqsABCDE* expression. Specifically, deletion of *pilY1* and *lasR* reduced *pqsA* promoter activity, AQ production, and surface-induced virulence. Meanwhile, constitutive expression of *pqsABCDE* was sufficient to restore the surface virulence of *pilY1* and *lasR* mutants. Epistasis studies on both virulence and AQ production revealed that the repression of AlgR by PilY1 promotes high levels of *pqsABCDE* expression in surface-attached cells. A similar epistatic relationship between PilY1 and the AlgR-AlgZ system was recently implicated in the virulence of *P. aeruginosa* towards *C. elegans* hosts [34], suggesting that this pathway is also important in other virulence contexts. We extend these results by showing that AlgR acts indirectly on the *pqsABCDE* operon to influence expression. Repression of AlgR by PilY1 upregulates the transcription factor RhlR, which then induces *pqsABCDE* expression through an upstream promoter. RhlR was previously shown to repress *pqsABCDE* in other contexts [13], such that in different conditions RhlR appears capable of either inducing or repressing *pqsABCDE*. The determinants of this sign difference in RhlR regulation will be an important topic of future investigation. For example, under some conditions, expression of Lrs1 by RhlR might disrupt the downstream *pqsABCDE* expression by preventing PqsR from accessing the binding site at the *pqsA* promoter, while in other conditions transcription from the Lrs1 promoter can run through the Lrs1 termination sequence to induce *pqsABCDE*. In any event, our findings provide additional complexity to the link between the Rhl and PQS systems, underscoring the interconnected relationship between the multiple QS signaling systems of *P. aeruginosa* and the ability of other stimuli to intersect with these systems.

The fact that AQs are sufficient to explain surface-induced *D. discoideum* killing was surprising because *P. aeruginosa* is generally thought to kill hosts using a large and redundant set of cytotoxins [4,35]. The large number of toxins present in *P. aeruginosa* could be due to the preferential ability of different toxins to kill different host cell types. For example, surface-dependent virulence towards mammalian monocytes was only partially disrupted by loss of AQ production, suggesting that other known virulence factors like the type-III secretion system (T3SS) work with AQs to kill these cells. Much as mutants lacking AQs retain partial virulence towards monocytes, T3SS mutants retain partial virulence in multiple infection systems [36,37], which is consistent with these toxins working in parallel to target hosts that have more complex systems for pathogen defense. While future studies will be needed to dissect the specific contributions of different virulence factors in different contexts, our study highlights the value of quantitative assays that can define specific capacities.

Could AQs act as cytotoxins in human infections? HHQ and other AQs are present in the serum, urine, and sputum of CF patients with *P. aeruginosa* infections [38,39], and have been shown to correlate with clinical progression [38,39]. Quantitative reports of AQ levels in human infections are limited [40]. While available values from clinical samples and animal infection models often vary and depend on the infection site, reports suggest values in CF sputum are in the mid to low nM range [19]. While these levels do not approach the levels of purified AQs necessary for cytotoxicity in our studies, our findings that bacterially produced AQs are more potent than the purified compound, which could result from their packaging in secreted OMVs, suggest that the activity of these molecules depend highly on the context in which they are made and how they are delivered to cells. These findings thus warrant a

reevaluation of the importance of AQs as direct cytotoxins in different animal infection models. Furthermore, given the chronic nature of many *P. aeruginosa* infections [2,35], and correlations between AQ levels and clinical progression [39,40], prolonged exposure to sub-µM levels of AQs could significantly impact tissue health over time.

What are the predominant AQ species present during *P. aeruginosa* infection? The conversion of HHQ to downstream AQs requires oxygen [41], and many infection sites are microaerophilic [42,43]. Indeed, HHQ levels were found to be higher than PQS or HQNO in a mouse burn infection model [26], where HHQ concentrations reached 0.6 µg mg$^{-1}$ in affected tissues [26]. Since bacterial biofilms on surfaces such as the CF lung are also oxygen-limited [42,43], HHQ may warrant particular attention as a candidate AQ cytotoxin in these conditions. Although low-oxygen conditions favor HHQ production over other AQs, total AQ levels may be lower in these conditions [41], such that further studies of the interplay of oxygen and surface sensing are needed to fully untangle these pathways. Furthermore, HHQ is a poor agonist of PqsR, with weaker activation than PQS [26] (also see Fig 2A). Consequently, blocking the conversion of HHQ to PQS could represent a mechanism for *P. aeruginosa* to specifically induce a cytotoxic AQ without inducing competing PQS-dependent targets such as pyocyanin. Given that PQS is the most potent PqsR agonist among AQs [26] and that a recent study found that HQNO is the most potent antibiotic [44], we suggest that AQs could be functionally specialized with PQS serving primarily as a QS signaling molecule, HQNO primarily for interbacterial competition, and HHQ for host cell cytotoxicity under surface-associated oxygen-limited conditions. Finally, we note that the different activities of AQs are not mutually exclusive such that induction of AQs upon surface association could represent a powerful strategy to simultaneously initiate cytotoxicity to ward off engulfment by phagocytic cells, suppress immune function [17], and signal additional downstream factors to promote factors associated with later stages of infection [14,18].

## Materials and Methods

### Bacterial strains, plasmids, and growth conditions

The strains and plasmids used in this study are described in S1 Table and S2 Table. Bacterial cultures were routinely grown in lysogeny broth (LB) broth at 37˚C with aeration or on LB solidified with 1.5% agar (BD Biosciences, San Jose, CA). When stated, bacteria were grown in PS:DB media, which consists of development buffer (DB) (5 mM KH$_2$PO$_4$, 5 mM Na$_2$HPO$_4$, 2 mM MgCl$_2$, 1 mM CaCl$_2$ pH 6.5) and 10% (v/v) PS medium (10 g L$^{-1}$ Special Peptone (Oxoid, Hampshire, United Kingdom), 7 g L$^{-1}$ Yeast Extract (Oxoid, Hampshire, United Kindom), 10 mM KH$_2$PO$_4$, 0.45 mM Na$_2$HPO$_4$, 15 g L$^{-1}$ glucose, 20 nM vitamin B12, 180 nM Folic Acid, pH 6.5). Antibiotics were added at the following concentrations when appropriate: carbenicillin 300 µg mL$^{-1}$, gentamycin 30 µg mL$^{-1}$, and tetracycline 200 µg mL$^{-1}$ for *P. aeruginosa*; 100 µg mL$^{-1}$, gentamycin 30 µg mL$^{-1}$, and tetracycline 15 µg mL$^{-1}$ for *E. coli*. Expression of P$_{tac}$- or P$_{lac}$-controlled genes was induced with 1 mM IPTG. When indicated, cultures were supplemented with HHQ, PQS, or HQNO (Cayman Chemicals, Ann Arbor, MI). Unless otherwise stated, chemicals and reagents were purchased from Sigma Aldrich (St. Louis, MO).

### Strain construction

Primers used in this study are described in S3 Table. All gene deletions described here are unmarked, in-frame deletions generated by two-step allelic exchange, as described previously [45]. Briefly, upstream and downstream homology arms flanking the relevant gene were amplified with primer pairs (-KO1,-KO2 and -KO3,-KO4; S3 Table), fused through overlap extension PCR (OE-PCR), and cloned into restriction sites of plasmid pEXG2. The pEXG2 plasmid was

integrated into *P. aeruginosa* through conjugation using the donor strain *E. coli* S17. Exconjugants were selected on gentamycin and then the mutants of interest were counter-selected on 5% sucrose. Transposon insertions obtained from the PA14 Transposon Mutant Database [46] were transferred between strains using the lambda-Red recombination system [47].

To generate the *pqsABCDE* overexpression strain, the $P_{OXB20}$ promoter was amplified from the plasmid pSF-OXB20 (Oxford Genetics, Cambridge, MA) using primer pair (OXB20-5 and OXB20-3) and spliced between two 400 bp fragments that flank the *pqsA* promoter, which were amplified from gDNA using primers (pqsUP-5, pqsUp-3) and (pqsDOWN-5, pqsDOWN-3), respectively. The resulting construct was cloned into pEXG2 and integrated onto the chromosome using allelic exchange. Strains expressing *pqsABCDE* under the control of $P_{OXB11}$ and $P_{OXB15}$ were made similarly using the same primer set for promoter amplification. To generate the inducible *pqsR* expression construct, the *pqsR* gene was amplified from gDNA with primer pair (pqsR-5, pqsR-3) and cloned into pUC18-mini-Tn7T-LAC. Proper gene orientation was confirmed by restriction mapping, and the resulting construct was integrated onto the chromosome by co-electroporation with pTNS2 using methods described previously [48]. To generate the *algRD54E* mutation and overexpression construct, the *algR* gene was amplified from gDNA using primers (algR-pUC-5, algR-pUC-3) and subcloned into pUC19 (New England Biolabs, Ipswich, MA). Site-directed mutagenesis was performed by amplification of pUC19::*algR* with primers (algR-D54E-Fw, algR-D54E-Rv), and the mutant allele was either cloned into pEXG2 and integrated into the *P. aeruginosa* chromosome by allelic exchange, or cloned into pBBRMC3 to obtain the overexpression construct. The $P_{pqsA}-m$-*Cherry* promoter fusion construct was generated by amplifying an approximately 500 bp fragment upstream of the *pqsA* gene with primer pair (PpqsA-1, PpqsA-2) and fusing it by OE-PCR to a promoterless *mcherry* gene, amplified from mini-CTX-2::PA1/04/03-mCherry with primer pair (PpqsA-3, PpqsA-4). The resulting fragment was cloned into the mini-CTX-2 plasmid and integrated into the chromosome at the CTX-2 phage attachment site (*attB*). The *Plrs1-mCherry* reporter was constructed similarly using primer pairs (Plrs1-1, Plrs1-2) and (Plrs1-3, Plrs1-4). To generate the fluorescent AQ biosensor, a fragment containing the *pqsA* promoter and PqsR binding site (-19 bp to -219 bp) was amplified from gDNA using primer pair (pqsA-PaQa-5, pqsA-PaQa-3), and cloned into the BamHI/XhoI sites of pUCP18:: $P_{pqsA}-YFP\ P_{rpoD}-mKate$. This plasmid was then transformed into a PAO1 strain containing a deletion in *pqsA*, and replacement of the *pqsR* promoter with a constitutive $P_{tac}$ promoter. We note that while our virulence assays were all performed with the PA14 strain of *P. aeruginosa*, we used the PAO1 strain for the biosensor since the reporter is avirulent and does not interfere with PA14 virulence while PAO1 maintains plasmids at higher levels than PA14.

Strains producing fluorescently labeled outer membrane vesicles (OMVs) express the construct $P_{OXB20}$-*oprM*::*mNeonGreen*. To generate this construct, the N-terminal region of the *oprM* gene was amplified from *P. aeruginosa* gDNA using primers (oprM-Neon-1, oprM-Neon-2) and fused by OE-PCR to the *mNeonGreen* gene, which was amplified using primers (oprM-Neon-3, oprM-Neon-4). The resulting fragment was cloned into the NotI/XhoI sites of plasmid pSF-OXB20. The $P_{OXB20}$-*oprM*::*mNeonGreen* sequence was then amplified with primers (oprM-Neon-5, oprM-Neon-6) and cloned by Gibson Assembly into plasmid pUCP18 linearized with HindIII and PciI.

### *D. discoideum* and mammalian cell culture

*D. discoideum* AX3 was maintained axenically as described previously [5,49]. Briefly, frozen stocks were inoculated into overnight cultures of *E. coli* B/r, and plated on GYP plates. After incubation for 4–6 days at 22°C, individual spores were inoculated into PS media

supplemented with Antibiotic-Antimycotic (AA) solution, and incubated at 22°C 100 rpm. When cultures reached approximately $1 \times 10^6$ cells $mL^{-1}$, cells were back-diluted 1:100 in fresh PS media. Axenic cultures were maintained for up to 1 month. J774A.1 mouse monocytes (ATCC TIB-67) and A549 human lung epithelial cells (ATCC CCL-185) were grown at 37°C with 5% $CO_2$ in Dulbecco's Modified Eagle's Medium (Gibco, Dublin, Ireland) supplemented with 10% fetal bovine serum and Penicillin-Streptomycin solution (Invitrogen, Grand Island, NY). Cells were passaged according to the ATCC protocols.

### *D. discoideum* cell death assay

Cell death assays were performed as described previously with minor modifications [5]. Overnight cultures of *P. aeruginosa* were diluted 1:100 in PS:DB media and grown to OD = 0.6–0.8 at 37°C with aeration. Cultures were transferred to glass-bottom dishes (Mattek, Ashland, MA) and incubated for an additional 1 hour at 100 rpm on a rotary shaker. For the planktonic condition, aliquots of culture media were washed with PS:DB, concentrated 20-fold, and plated onto fresh glass-bottom dishes. For the surface-attached condition, culture media was aspirated and surface-attached cells were washed with PS:DB. Aliquots of *D. discoideum* culture (between $2–5 \times 10^5$ cells $mL^{-1}$) were washed with PS:DB, and added to planktonic and surface-attached bacteria to achieve the appropriate multiplicity of infection (MOI), which ranged between roughly 500:1 to 1000:1 (*P. aeruginosa* to *D. discoideum*), unless otherwise stated. See Siryaporn et al. [5] for details regarding assay validation and MOI quantification. The combined samples were covered with a 1% agar pad, prepared by pouring molten 1% agar in PS:DB containing 1 μM calcein-AM (Invitrogen, Grand Island, NY) on a glass surface, and cutting the solidified pad into 1 cm x 1 cm sections. Samples were analyzed by imaging cells with phase contrast and FITC channels after 1 hour of incubation at 25°C using fluorescence microscopy. Cell death was quantified by counting the total number of calcein-AM-positive and -negative cells. All reported values of percent cell death are averages of at least three independent experiments (biological replicates). Each set of experiments included wild type and *pqsA* mutant controls, and each measurement was of 150–500 cells.

For quantification of *D. discoideum* killing by *P. aeruginosa* attached to agar or agarose surfaces, *P. aeruginosa* cultures (OD = 0.6) were transferred to 3 cm petri dishes coated with solidified agar or agarose prepared in PS:DB media at the indicated concentration, and cultures were grown for 1 h at 37°C with shaking (100 rpm). Pads were excised and inverted onto *D. discoideum* samples mixed with calcein-AM, and imaged as described above. For the PDMS condition, Sylgard-527 was prepared, degassed for 30 min, used to coat 3 cm petri dishes, and cured for at least 24 h at room temperature. Virulence assays for plastic and PDMS-coated plates were performed as described above for glass surfaces.

For quantifying cytotoxicity of purified AQs under conditions of the microscopy-based cell death assay (S7 Fig), AQs were diluted 1:200 into molten 1% agar in PS:DB and pads were prepared as described above. Samples of *D. discoideum* were transferred to glass-bottom dishes and covered with a 1 cm x 1 cm agar pad and incubated at 25°C for 1 h. For quantifying cytotoxicity of purified AQs in axenic *D. discoideum* cultures (Fig 1D), *D. discoideum* was subcultured to 10,000 cells $mL^{-1}$ in fresh PS media with antibiotics and varying concentrations of AQs. Cultures were incubated at 22°C with shaking at 450 rpm, and cell density was measured by counting with a hemocytometer. Experiments were performed with five biological replicates.

### TIB-67 cytotoxicity assay with purified AQs

TIB-67 monocytes cells were seeded at a density of 150,000 cells or 75,000 cells $cm^{-1}$ in a 96-well plate and incubated at 37°C 5% $CO_2$ for 24 h. Cells were treated with purified AQ

compounds dissolved in DMSO such that the final concentration of DMSO was less than 0.5% After 48 h, media was aspirated and the WST-8 reagent EZQuant (Alstem Bio, Richmond, CA) was used to assess cell viability. Values reported are averages of three biological replicates. At least three independent experiments were performed, and the trends observed in Fig 1E was observed across all experiments.

## Microscopy-based TIB-67 cell-death assay

For the microscopy-based assay, TIB-67 monocytes were seeded at a density of 75,000 cells cm$^{-2}$ and incubated at 37˚C 5% $CO_2$ for 24 h. TIB-67 monolayers were washed twice with phosphate buffered saline (PBS) (Gibco, Dublin, Ireland) and combined with *P. aeruginosa* samples. To prepared *P. aeruginosa* samples, cultures were grown following the procedures of the *D. discoideum* cell death assay. When cultures reached OD = 0.5–0.6, cultures were transferred to petri dishes coated with a thin layer of 1% agar in PS:PBS (10% (v/v) PS media in PBS with 1 mM $MgSO_4$ and 0.1 mM $CaCl_2$, pH = 7.2) and grown for an additional 1 h at 37˚C on a rotary shaker (100 rpm) to allow surface attachment. Surface-attached bacteria were washed twice with PS:PBS. The density of surface-attached bacteria could be adjusted based on force applied during washing steps using an automated pipette, and density was optimized to achieve a final multiplicity of infection (MOI) of 50:1 to 150:1 (*P. aeruginosa* to TIB-67). Propidium iodide (PI; 1 μM final) and sub-MIC dose of tetracycline (5 μg mL$^{-1}$ final) was added to prepared monocyte samples. Agar pads were excised and inverted onto prepared monolayers of TIB-67 monocytes. Concentrations of PI and tetracycline were calculated based on the volume of the agar pad. Samples were incubated at 30˚C in an incubated chamber, and cells were tracked over the course of 24 h using fluorescence microscopy. Cell death was monitored by PI staining of the DNA of non-viable cells. Bacterial co-culture with monocytes resulted in monocyte lysis (see S5 Fig) and subsequent diffusion of the nuclear PI stain. Therefore, *P. aeruginosa* virulence was most precisely quantified by counting viable monocytes (PI-negative) as opposed to dead monocytes (PI-positive). PI-negative cells were counted at various timepoints and divided by the number of PI-negative cells at 1 h in the same field of view.

## Plate-based AQ biosensor assay

Overnight cultures of the AQ biosensor was diluted 1:100 in LB with 1 mM IPTG and grown to late exponential phase (OD = 1.0–2.0). Bacteria were resuspended in fresh LB to OD = 1.0 and added to equal volumes of LB supplemented with various concentrations of AQs (purchased from Sigma-Aldrich) in a 96-well plate. Plates were sealed with a breathable membrane (DivBio, Dedham, MA) and incubated at 37˚C 450 rpm while monitoring fluorescence in the YFP (500 nm excitation/540 nm emission) and mKate (590 nm excitation/ 645 nm emission) channels. Biosensor signal was calculated by normalizing the YFP signal by the mKate signal and subtracting the baseline expression from a DMSO control. Six technical replicates were performed for all conditions shown in Fig 2A.

## Biosensor-based AQ quantification of surface-attached bacteria

Overnight cultures of *pqsH* mutant *P. aeruginosa* were grown following the procedures of the *D. discoideum* cell death assay. The AQ biosensor was prepared according to the procedures of the plate-based assay, but resuspended to OD = 0.4 in PS:DB. For planktonic samples, biosensor was added to equal volumes of planktonic cell suspensions. For surface-attached samples, biosensor was added to washed, surface-attached cells. Cells were covered with a 1% agar pad in PS:DB (1 mM IPTG), and incubated at 30˚C for 1.5 h. YFP and mKate fluorescence was measured by fluorescence microscopy, and biosensor signal was calculated by dividing the

YFP signal by the mKate signal, and then subtracting the baseline value of YFP/mKate signal, which was calculated by doping biosensor into a lawn of surface-attached Δ*pqsA P. aeruginosa*. For the quantification of HHQ levels in surface attached and planktonic *pqsH P. aeruginosa* (Fig 2D and 2E), mean biosensor signal was calculated for three biological replicates of each condition, and >7,500 cells were analyzed for each replicate and condition. For quantification of biosensor signal in *algR* and *pilY1* mutant strains, biosensor signal was measured after incubating for 2 h at 25°C following addition of the agar pad. The reported biosensor signal measurements for this experiment are averages from three biological replicates, and approximately 500 individual cells were analyzed for each replicate.

To analyze AQ levels in *P. aeruginosa* populations attached to monolayers of A549 human lung epithelial cells, monolayers were first prepared by seeding cells at a density of 20,000 cells cm$^{-2}$ in glass bottom dishes and incubating cells at 37°C 5% $CO_2$ for approximately 48 h. Overnight cultures of wild type and *pqsA* mutant *P. aeruginosa* were sub-cultured 1:100 in DMEM and grown at 37°C with aeration to OD = 0.6. Cell monolayers were incubated with DPBS for 5 min to remove residual antibiotics, and bacterial cultures were transferred to dishes containing monolayers, and incubated for 1 h at 37°C with shaking (80 rpm). Culture media was aspirated and monolayer-attached bacteria were washed twice with DPBS. Biosensor cells were prepared as described above, added to monolayer-attached bacteria, and covered in a 1% agar pad prepared with DMEM (with 1 mM IPTG; without phenol-red). Biosensor signal was measured by fluorescence microscopy after 1.5 h incubation at 30°C. For the planktonic condition, bacterial cultures at OD = 0.6 were grown for an additional h at 37°C with aeration, then bacteria were washed twice with DPBS, mixed with biosensor cells and transferred to fresh monolayers. To accurately assess the levels of bacterial attachment to A549 cells, which is not feasible using phase contrast imaging due to sample complexity, preliminary experiments were performed with *P. aeruginosa* expressing *oprM*::*mNeonGreen* (Fig 2F and S8 Fig). Fluorescent bacteria were not used for quantification of biosensor signal in Fig 2G due to the possibility of interference with the YFP signal. When indicated, monolayers were stained after aspiration of bacterial cultures by incubating cells with 5 μg mL$^{-1}$ Hoescht 33342 and/or 1 μg mL$^{-1}$ FM 4–64 in DPBS for 5 min prior to the two washing steps. Stains were not used for quantitative experiments in Fig 2G.

An HHQ standard curve (Fig 2C) was generated and used to convert biosensor signal to HHQ concentration. Biosensor was doped 1:50 into surface-attached Δ*pqsA P. aeruginosa* and covered with 1% agar pads made with PS:DB, supplemented with various concentrations of HHQ and 1 mM IPTG. Biosensor signal in response to HHQ standards was calculated as described above. A new standard curve was constructed for each independent experiment. All measurements of biosensor signal of surface-attached Δ*pqsH* fell within the working range of the HHQ standard curve.

## OMV isolation, quantification, and cytotoxicity measurements

Outer membrane vesicles (OMV) were isolated as described previously [27,50]. Because OMVs are produced at relatively low levels in the absence of AQs, we used the approach of MacDonald et al. [7] to stimulate OMV production in the *pqsA* mutant, and kept conditions the same for wild type cultures to minimize differences between samples. Briefly, overnight cultures of *P. aeruginosa* were subcultured 1:200 in LB and grown at 37°C 250 rpm to OD = 0.3–0.4. Polymyxin B was added to a final concentration of 4 μg mL$^{-1}$, and cultures were grown at 37°C 250 rpm until they reached an OD between 1.0–1.5. Culture supernatants were filtered through a 0.45 μm filter (Millpore, Burlington, MA), then centrifuged at 40,000 x g for 1 h. Pellets were resuspended in PBS, filtered through a 0.45 μm filter, and centrifuged for an

additional 1 h at 100,000 x g. Samples were resuspended in PBS and concentrated with a 100,000 MWCO centrifugal filter device (Millipore, Burlington, MA). Filtrate was used as a vehicle control. OMV concentrations were measured by addition of the lipophilic dye FM-464 (Fisher Scientific, Hampton, NH) and measuring fluorescence (506 nm excitation/ 750 nm emission).

For treatment of *D. discoideum* and TIB-67 monocytes with OMV samples, the OMV concentration of the wild type sample was adjusted to match the concentration of the *ΔpqsA* sample. Final OMV stocks were approximately 5000-fold concentrated compared to *ΔpqsA* culture supernatant. Stocks were diluted 1:200 (based on the volume of the agar pad) into each sample of TIB-67 monocytes. To lyse OMVs, SDS was added to concentrated stocks to a final concentration of 0.05% SDS and samples were incubated at 37˚C for 1 h. Lysis was confirmed by imaging aliquots of SDS-treated and untreated, fluorescently labelled OMVs that were isolated from *P. aeruginosa* expressing $P_{OXB20}$-*oprM*::*mNeonGreen*. To analyze the cytotoxicity of lysed and unlysed OMVs towards *D. discoideum*, OMV stocks were diluted 1:2500 into *D. discoideum* cultures growing axenically in PS:DB with 1 μM calcein-AM and incubated statically at 25˚C. SDS was added to the unlysed sample to match the final concentrated of the lysed sample ($2\times10^{-5}$% SDS) to ensure the presence of SDS does not influence the results of the assay. Viability based on calcein-AM staining was measured at 2 h using fluorescence microscopy.

## Quantification of fluorescent reporter expression

Reporter strains were grown according to the procedures of the *D. discoideum* cell death assay. Bacterial cultures were transferred to glass-bottom dishes when OD reached 0.6, and incubated at 37˚C with shaking (100 rpm) for 1 h. Cells were washed and isolated as described above, and single cells were imaged immediately after addition of the 1% agar pad using fluorescence microscopy. For measurements of the normalized *pqsA* promoter activity (Fig 4C), mean fluorescence intensity per cell was computed for 500–1000 cells, and average values were normalized by the expression of a constitutive $P_{tac}$−*mCherry* reporter, which was measured in the same manner as the $P_{pqsA}$ reporters. For measurements of *lrs1* promoter activity, mean fluorescence intensity per cell was computed for 500–1000 cells. All reported values are averages from at least three independent experiments (biological replicates).

## LC/MS-based quantification of AQ production

Overnight cultures of *P. aeruginosa* were subcultured 1:100 into 20 mL PS:DB and grown to OD = 1.5 at 37˚C. Cultures were extracted with equal volumes of ethyl acetate, dried and resuspended in methanol. Samples were analyzed by HPLC-MS on a 1260 Infinity Series HPLC system (Agilent, Santa Clara, CA) equipped with an automated liquid sampler, a diode array detector, and a 6120 Series ESI mass spectrometer using an analytical Luna C18 column (5 m, 4.6 x 100 mm, Phenomenex, Torrance, CA) operating at 0.6 mL min$^{-1}$ with a gradient of 25% MeCN in $H_2O$ to 100% MeCN over 18 min. A standard curve constructed from commercial AQ standards was used to calculated AQ concentrations in cultures.

## RNA isolation and qRT-PCR analysis of transcript abundance

Overnight cultures of *P. aeruginosa* were diluted 1:100 in PS:DB media and grown to OD = 0.6 at 37˚C with aeration. Cultures were transferred to 24 x 24 cm bioassay plates (Fisher Scientific, Hampton, NH) and incubated for 1 h at 37˚C with shaking at 100 rpm. For the planktonic condition, aliquots of culture media were collected and centrifuged to isolate bacteria. For the surface-attached condition, residual culture media was aspirated, and surface-attached cells were washed twice with PBS. Bacteria were removed from the surface using a cell scraper,

collected in PBS, and centrifuged to isolate bacteria. Bacterial pellets were resuspended in RNAprotect bacteria reagent (Qiagen, Hilden, Germany). Total RNA was isolated using miR-NEasy kit (Qiagen) with on column DNAse I digestion (Quaigen). Total RNA was converted to cDNA using SuperScript III reverse transcriptase and random primers (Fisher Scientific, Hampton, NH). Quantitative PCR (qPCR) was performed using SYBR-green qPCR Master Mix reagent (Fisher Scientific, Hampton, NH) on surface-attached and planktonic cDNA templates with probes for the *pqsA* gene, the intergenic region between *lrs1* and *pqsA* genes, and the gene encoding glyceraldehyde-3-phosphate dehydrogenase (*gapA*) as a control. Samples of cDNA were isolated from three independent biological replicates and results shown in Fig 4G are the average ΔΔCt value, calculated as described previously [51], between surface-attached and planktonic samples, using *gapA* as a housekeeping control (n = 3).

### Image processing

To process images of biosensor cells doped into populations of *P. aeruginosa* for presentation (Fig 2), YFP and mKate fluorescence images were background subtracted, and the constitutive mKate signal was used to segment individual biosensor cells and construct a mask that was applied to the YFP channel. The YFP channel was then divided by the mKate channel to produce biosensor signal. The black background represents the absence of a cell and has a value of 0. Color scales for biosensor signal are shown each time a biosensor image is reported. Unless otherwise stated, fluorescence images were background subtracted to allow improved comparisons of relative fluorescent signal of individual cells and image clarity.

### Supporting information

**S1 Movie. Time-lapse imaging of *D. discoideum* infected with planktonic *P. aeruginosa*.** Planktonic bacteria were isolated following the procedures of the *D. discoideum* cell-death assay (see Materials and Methods) and added to *D. discoideum* at an MOI of approximately 50:1 (bacteria:amoeba). Samples were covered with a 1.5% agar pad and incubated at 25˚C. Phase contrast images were taken every 5 min for 200 min.
(AVI)

**S2 Movie. Time-lapse imaging of *D. discoideum* infected with surface-attached *P. aeruginosa*.** Surface-attached bacteria were isolated following the procedures of the *D. discoideum* cell-death assay (see Materials and Methods) and added to *D. discoideum* at an MOI of approximately 50:1 (bacteria:amoeba). Samples were covered with a 1.5% agar pad and incubated at 25˚C. Phase contrast images were taken every 5 min for 200 min.
(AVI)

**S1 Fig. *P. aeruginosa* virulence following attachment to surfaces of various composition and stiffness.** Quantification of *D. discoideum* killing by planktonic and surface-attached *P. aeruginosa* attached to substrates of various composition and stiffness after 1 h of co-culture. Cell death was indicated by positive staining by the fluorescent dye calcein-AM. Values are averages of three biological replicates and error bars represent standard error. Approximately 250–300 cells were analyzed for each measurement. PDMS is Sylgard-527. Values for the planktonic and glass condition are those in Fig 4B.
(TIF)

**S2 Fig. Effect of known virulence regulators on surface-induced virulence.** Quantification of *D. discoideum* killing by surface-attached *P. aeruginosa* mutants after 1 h of co-culture. Cell death was indicated by positive staining by the fluorescent dye calcein-AM. Values are averages of three independent experiments and error bars represent standard error. Approximately

150–300 cells were analyzed for each measurement.
(TIF)

**S3 Fig. Effect of multiplicity of infection (MOI) on *pqsA*-dependent virulence.** Quantification of *D. discoideum* killing by surface-attached wild type and *pqsA* mutant *P. aeruginosa* at varying initial MOI (*P. aeruginosa* to *D. discoideum*). (A) Representative images of *P. aeruginosa* mixed with *D. discoideum* at the labelled MOI at t = 0 h (scale bars = 50 μm). (B) Quantification of *D. discoideum* death after 5 h. Values are averages of three biological replicates and error bars represent standard error. Approximately 250–300 cells were analyzed for each measurement. Representative images of *D. discoideum* treated with *pqsA* mutant bacteria at varying MOI, demonstrating near-complete phagocytosis of bacterial lawns after 5 h when initial MOI is at or below 250:1 (scale bars = 150 μm).
(TIF)

**S4 Fig. Effect of constitutive-promoter strength on the rescue of *pqsH* virulence.** Quantification of *D. discoideum* killing by surface-attached *P. aeruginosa pqsH* mutants after 1 h of coculture. Expression of *pqsABCDE* genes are controlled by a constitutive promoter with high ($P_{OXB20}$), moderate ($P_{OXB15}$), or low ($P_{OXB11}$) levels of expression. Cell death was indicated by positive staining by the fluorescent dye calcein-AM. Data for wild type and *pqsH* are from Fig 1C (n = 3). Other values shown are the average of two independent experiments and error bars represent standard deviation. Approximately 300–500 cells were analyzed for each measurement.
(TIF)

**S5 Fig. Microscopy-based monocyte virulence assay.** (A) Schematic of the monocyte cell-death assay described in *Materials and Methods*. (B) Representative images of TIB-67 monocytes treated with surface-attached wild type *P. aeruginosa* at a MOI of approximately 50:1 (bacteria:amoeba) after 1 and 15 h of incubation at 30˚C. Cell death is indicated by propidium iodide staining.
(TIF)

**S6 Fig. AQ-dependent monocyte killing.** Representative images of TIB-67 monocytes co-cultured with surface-attached *P. aeruginosa* or treated with exogenous HHQ under conditions of the microscopy-based virulence assay (t = 16 h). Propidium iodide (PI) staining of DNA of non-viable cells indicates cell death (scale bars = 50 μm).
(TIF)

**S7 Fig. Rapid killing of *D. discoideum* by purified HHQ.** Cytotoxicity of purified HHQ towards *D. discoideum* under conditions of the *P. aeruginosa* virulence assay. HHQ was added to 1% agar pads used for imaging. Values are mean ± SEM of three biological replicates (n = 3). Approximately 200–300 cells were analyzed for each measurement.
(TIF)

**S8 Fig. Quantification of HHQ in strains constitutively expressing *pqsABCDE* at different levels.** AQ biosensor-based quantification of HHQ levels in surface-attached *pqsH P. aeruginosa* populations. Expression of *pqsABCDE* genes are controlled by a constitutive promoter with high ($P_{OXB20}$), moderate ($P_{OXB15}$), or low ($P_{OXB11}$) levels of expression. Mean biosensor signal was calculated from >2,500 cells and converted to HHQ levels using a standard curve constructed using purified HHQ standards.
(TIF)

**S9 Fig. Attachment of *P. aeruginosa* to monolayers of A549 human lung epithelial cells.** Confirmation of bacterial attachment to monolayers of A549 human lung epithelial cells. Wild type *P. aeruginosa* expressing constitutive *oprM*::*mNeonGreen* was grown to OD = 0.6 in DMEM and transferred to confluent monolayers of A549 cells. Co-cultures were grown for 1 h at 37°C with shaking (80 rpm), then monolayers were washed twice with DPBS, treated briefly with FM 4–64 and Hoescht, then covered with 1% agar pad prepared with PBS. Fluorescent images were taken immediately (scale bars = 100 μm)
(TIF)

**S10 Fig. Isolation and lysis of fluorescently tagged outer membrane vesicles (OMV).** Isolation and lysis of fluorescently labelled outer membrane vesicles (OMVs) from *P. aeruginosa*. (A) Representative images of $P_{OXB20}$-*oprM*::*mNeonGreen* *P. aeruginosa* cells and OMVs isolated from supernatants of this strain following procedures described in *Materials and Methods*. OMVs were treated with 0.05% SDS at 37°C for 1 h to induce OMV lysis and subsequent diffusion of the fluorescent signal. SDS concentrations lower than 0.05% were not sufficient to cause lysis. Fluorescent signal does not accurately reflect OMV size (generally < 200 nm diameter) due to the diffraction limit of light microscopy (scale bars = 20 μm) (B) Representative measurements of OMV levels in samples isolated from WT and *pqsA P. aeruginosa* using FM464-based quantification method described in *Materials and Methods*. FM464 signal is fluorescence at (ex = 515 em = 645) in arbitrary fluorescence units. Filtrate is eluent obtained from concentrating OMV samples in 100 MWCO centrifugal concentration units. (C) Protein content of WT and *pqsA* OMVs analyzed by SDS-PAGE and Coomasie staining.
(TIF)

**S11 Fig. Effect of OMV lysis on AQ-based biosensor stimulation.** Treatment of AQ biosensor with lysed and unlysed OMV samples isolated from WT *P. aeruginosa*. OMV stocks isolated from non-fluorescent bacteria were added to biosensor cells in 96-well format, and YFP and mKate fluorescence were measured after 2 h incubation at 37°C 450 rpm. YFP/mKate fluorescence was subtracted by the mean biosensor signal of a sample treated with unlysed OMVs isolated from a *pqsA* mutant. Values shown are mean and error bars represent SD (n = 5). Dilution show is within the linear range of the AQ biosensor as shown in Fig 2A.
(TIF)

**S12 Fig. AQ quantification in populations of surface-attached *pilY1* and *algR* mutant bacteria.** Biosensor-based quantification of AQ levels in surface-attached *P. aeruginosa* populations. (A) Representative images of AQ biosensor doped (1:100) into *P. aeruginosa* samples as described in Fig 2 (scale bars = 10 μm). (B) Quantification of biosensor signal described in (A). Mean YFP/mKate fluorescence intensity per cell was calculated for approximately 500 cells, and values were baseline subtracted by the mean YFP/mKate fluorescence of biosensor cells doped into the surface-attached *pqsA* mutant. Values are averages of three independent experiments and error bars represent standard error.
(TIF)

**S1 Table. Bacterial strains and cell lines used in this study.**
(DOCX)

**S2 Table. Plasmids used in this study.**
(DOCX)

**S3 Table. Primers used in this study.**
(DOCX)

**S1 Data. Source data for primary and supplemental figures.**
(XLSX)

## Acknowledgments

We wish to thank members of the Gitai and Shaevitz labs for helpful discussions and comments on the manuscript, the Bassler and O'Toole labs for strains and reagents, and Ben Bratton for technical assistance and data analysis.

## Author Contributions

**Conceptualization:** Geoffrey D. Vrla, Zemer Gitai.

**Data curation:** Geoffrey D. Vrla.

**Formal analysis:** Geoffrey D. Vrla.

**Funding acquisition:** Mohammad R. Seyedsayamdost, Zemer Gitai.

**Investigation:** Geoffrey D. Vrla, Mark Esposito, Chen Zhang.

**Methodology:** Geoffrey D. Vrla, Mark Esposito, Chen Zhang, Mohammad R. Seyedsayamdost, Zemer Gitai.

**Project administration:** Zemer Gitai.

**Resources:** Yibin Kang, Mohammad R. Seyedsayamdost, Zemer Gitai.

**Supervision:** Zemer Gitai.

**Validation:** Geoffrey D. Vrla, Mark Esposito, Chen Zhang.

**Visualization:** Geoffrey D. Vrla.

**Writing – original draft:** Geoffrey D. Vrla, Zemer Gitai.

**Writing – review & editing:** Geoffrey D. Vrla, Zemer Gitai.

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
