## [Decision Letter · Decision Letter 0]

26 Jun 2020

Dear Dr. Gitai,

Thank you very much for submitting your manuscript "Cytotoxic alkyl-quinolones mediate surface-induced virulence in Pseudomonas aeruginosa" for consideration at PLOS Pathogens. As with all papers reviewed by the journal, your manuscript was reviewed by members of the editorial board and by several independent reviewers. In light of the reviews (below this email), we would like to invite the resubmission of a significantly-revised version that takes into account the reviewers' comments.

There is overall enthusiasm for the manuscript and all reviewers agree that it is well written and interesting. In addition to addressing the specific experimental concerns expressed by the reviewers, I ask that you explicitly address the issue of relevance of attachment to glass and the disconnect between MOI used in the dictyo versus monocyte infection studies as indicated by reviewer #1.

We cannot make any decision about publication until we have seen the revised manuscript and your response to the reviewers' comments. Your revised manuscript is also likely to be sent to reviewers for further evaluation.

Sincerely,

Matthew C Wolfgang

Associate Editor

PLOS Pathogens

Alan Hauser

Section Editor

PLOS Pathogens

Kasturi Haldar

Editor-in-Chief

PLOS Pathogens

orcid.org/0000-0001-5065-158X

Michael Malim

Editor-in-Chief

PLOS Pathogens

orcid.org/0000-0002-7699-2064

There is overall enthusiasm for the manuscript and all reviewers agree that it is well written and interesting. In addition to addressing the specific experimental concerns expressed by the reviewers, I ask that you explicitly address the issue of relevance of attachment to glass and the disconnect between MOI used in the dictyo versus monocyte infection studies as indicated by reviewer #1.

Reviewer's Responses to Questions

**Part I - Summary**

Reviewer #1: This is a well-written and technically sound manuscript that describes a mechanism of surface-dependent killing by P. aeruginosa. Challenging, however, is the situational context to which the findings apply and might translate to actual mechanisms of virulence. The authors propose that attachment of Pa to glass activates surface dependent killing and all of the reported findings are in the context of a glass surface. How is attachment to a glass surface relevant? Does attachment to plastic trigger the same response? What is the relevant in vivo attachment surface? It would have to be a host surface, either cellular or matrix. Does attachment to host surfaces trigger the same response?

Whereas the MOI for the Dictyo experiments was 500-1000, the monocyte experiments used an MOI of 0.02 – 0.007. It is hard to know whether an MOI of 500-1000 is physiologically relevant but there is some concern that killing ~90% of the Dictyo within 60 minutes (Fig. 1F) is an artefact of the MOI and in doing obscures more relevant mechanisms that function at a much lower (and possibly more relevant) MOI, such as the T3SS which can also contributes killing.

The cited study from the Iglewski group used also strain PA14 and THP-1 monocytes and observed nearly 100% lysis at 6 hours using an 0.1-1 MOI. Fig. 1F shows only 25% killing after 18 hrs. Does attachment to glass suppress T3SS-dependent killing?

Reviewer #2: The authors were very responsive to my comments. The revised manuscript now contains (again?) additional data on OMVs, demonstrating OMV trafficked AQs display enhanced cytotoxicity towards host cells, but do not enhance their efficacy as autoinducers. These additional experiments are also well executed and controlled. In conclusion, the revised manuscript is well-written and -presented, with clear and well controlled experiments that justify the conclusions drawn.

Reviewer #3: In this study Vrla et al. aimed to assess and identify the possible role of AQs produced by P. aeruginosa during infection, and they investigated the role of AQs production in response to the surface attachment. To do so, they used Dictyostelium and mammalian monocytes as infection/cytotoxicity model systems.

The authors showed a significant decrease in Dictyostelium cell death in pqsA and pqsR mutants, leading them to the finding that HHQ and PQS are responsible for both mammalian and Dictyostelium cells. However, HHQ is sufficient for cytotoxicity against Dictyostelium. The authors developed several bioreporters that confirm higher levels of expression of AQs in surface-attached cells compared to planktonic cells.

One of the central and exciting findings is the increased cytotoxicity of intact OMVs compared to the lysed OMVs and the imaging methods used.

I found the regulatory role of small RNA Lrs1 one of the main findings of this manuscript that connects two main QS regulatory systems in P. aeruginosa. However, this finding is a bit lost. Maybe more discussion on complicated QS system and it’s role in virulence during chronic infections and biofilms of P. aeruginosa can add more emphasis to this finding.

**Part II – Major Issues: Key Experiments Required for Acceptance**

Reviewer #1: (No Response)

Reviewer #2: N/A

Reviewer #3: (No Response)

**Part III – Minor Issues: Editorial and Data Presentation Modifications**

Reviewer #1: (No Response)

Reviewer #2: N/A

Reviewer #3: line 184: Not sure if the surface attachment leads to AQs level beyond reported previously, couldn't it be that when other virulence factors are available, small concentrations of HHQ is enough for the levels of cytotoxicity seen by whole cells?

Figure 2 A and D, can be moved to the supplementary documents.

Figure 4. E can be moved to the supplementary documents, this will allow more resolution and emphasis on sections C and G.

Despite the efforts on OMV isolation from planktonic cultures, it would be a valuable addition in follow up studies to isolate the OMVs from biofilms or surface attached cells,

PLOS authors have the option to publish the peer review history of their article (what does this mean?). If published, this will include your full peer review and any attached files.

Reviewer #1: No

Reviewer #2: No

Reviewer #3: No
---

## [Decision Letter · Decision Letter 1]

4 Aug 2020

Dear Dr. Gitai,

We are pleased to inform you that your manuscript 'Cytotoxic alkyl-quinolones mediate surface-induced virulence in Pseudomonas aeruginosa' has been provisionally accepted for publication in PLOS Pathogens.

Best regards,

Matthew C Wolfgang

Associate Editor

PLOS Pathogens

Alan Hauser

Section Editor

PLOS Pathogens

Kasturi Haldar

Editor-in-Chief

PLOS Pathogens

orcid.org/0000-0001-5065-158X

Michael Malim

Editor-in-Chief

PLOS Pathogens

orcid.org/0000-0002-7699-2064

Reviewer Comments (if any, and for reference):

Reviewer's Responses to Questions

**Part I - Summary**

Reviewer #1: NA

**Part II – Major Issues: Key Experiments Required for Acceptance**

Reviewer #1: NA

**Part III – Minor Issues: Editorial and Data Presentation Modifications**

Reviewer #1: NA

PLOS authors have the option to publish the peer review history of their article (what does this mean?). If published, this will include your full peer review and any attached files.

Reviewer #1: No

---

## [Editor Report · Acceptance letter]

9 Sep 2020

Dear Dr. Gitai,

We are delighted to inform you that your manuscript, "Cytotoxic alkyl-quinolones mediate surface-induced virulence in *Pseudomonas aeruginosa*," has been formally accepted for publication in PLOS Pathogens.

Best regards,

Kasturi Haldar

Editor-in-Chief

PLOS Pathogens

orcid.org/0000-0001-5065-158X

Michael Malim

Editor-in-Chief

PLOS Pathogens

orcid.org/0000-0002-7699-2064